# A Preliminary Characterisation of Innovative Semi-Flexible Composite Pavement Comprising Geopolymer Grout and Reclaimed Asphalt Planings

**DOI:** 10.3390/ma13163644

**Published:** 2020-08-17

**Authors:** An Thao Huynh, Bryan Magee, David Woodward

**Affiliations:** Belfast School of Architecture and the Built Environment, Ulster University, Belfast BT37 0QB, UK; huynh-a@ulster.ac.uk (A.T.H.); b.magee@ulster.ac.uk (B.M.)

**Keywords:** reclaimed asphalt planing, geopolymer grout, semi-flexible pavement, permeable porosity, compressive strength, ultrasonic pulse velocity

## Abstract

This article considers semi-flexible composite (SFC) pavement materials made with reclaimed asphalt planings (RAP) and geopolymer cement-based grouts. Geopolymer grouts were developed and used to fill the internal void structure of coarse RAP skeletons with varying levels of porosity. The geopolymer grouts were formulated at ambient temperature using industrial by-products to offer economic and environmental savings relative to conventional Portland cement-based grouting systems. They were characterised on flowability, setting time, and compressive strength. The effect of grout and RAP on SFC material performance was evaluated using permeable porosity, compressive strength, and ultrasonic pulse velocity. SFC performance was significantly influenced by both grout type and RAP content. Improved performance was associated with mixtures of high-flowability/high-strength grout and low RAP content. A practical limitation was identified for combination of grout with low-flowability/fast-setting time and well-compacted RAP skeletons. Solids content exceeding 49% by volume was not feasible, owing to inadequate grout penetration. A suite of SFC materials was produced offering performance levels for a range of practical pavement applications. Preliminary relationships enabling prediction of SFC elastic modulus based on strength and/or ultrasonic pulse velocity test data are given. A pavement design is given using SFC as a sub-base layer for an industrial hardstanding.

## 1. Introduction

Construction of highway pavement and hardstanding assets can consume significant amounts of natural resources such as aggregate, bitumen, and concrete, as well as energy in material heating, mixing, and compaction [1,2,3]. Significant quantities of greenhouse gases are emitted into the atmosphere through aggregate extraction and asphalt and Portland cement production [4,5]. As pressure to reduce natural resource extraction grows, using construction and industrial wastes as an alternative to raw materials can help to resolve environmental issues caused by depletion of natural sources and reduce wastes going to landfill. Construction products using cold recycling techniques to minimise use of energy and natural resources play an important part in the delivery of environmentally responsible infrastructure systems.

Recycling reclaimed asphalt planings (RAP) and other industrial wastes has drawn tremendous attention from researchers and scientists. Generated from road surfacing maintenance works or full-depth pavement removal and reconstruction, RAP has been the most important source of recycled material used in the pavement construction for many years [6]. It can be recycled into hot [7], warm [4,8], and cold mix asphalt [9], with up to 100% aggregate replacement levels possible depending on different design purposes. While the use of RAP as a construction product is potentially restricted due to a perception of lower strength and durability [10,11], research reports its use leading to increased stiffness levels compared to conventional hot mix asphalt (HMA) [12,13]. In addition to its reuse in asphalt, work has explored alternative uses of RAP by combining it with Portland cement [14,15,16] to create cementitious grouted materials. Generally referred to as semi-flexible composite (SFC) pavements [17,18], grouted macadam [15,19] or resin-modified pavement [20], their use has typically been for heavy and slow trafficked-areas such as distribution centres, industrial areas, or airports. Hossiney at al. [21] studied properties including compressive and flexural strength of Portland concrete containing up to 40% by volume of aggregate replaced by RAP, with performance generally decreasing with increasing RAP content. Laboratory test results by Huang et al. [16] indicated that the energy absorbing-toughness value of Portland concrete containing RAP improved compared to normal concrete with natural aggregate. This can be explained by the aged bitumen layer coating RAP behaving as an energy absorbing layer between the coarse aggregate and cement matrix, leading to reduced levels of crack propagation [22]. Commercial cement-based products [23] incorporating single-size open texture RAP with 25–30% voids and cement mortar have been developed to produce pavement materials with high load-bearing capacity and rapid installation times. Such examples of commercial products offer sustainable options for construction products because of their long-term, in-service performance abilities.

Against this background, reported in this paper is an investigation into the use of geopolymer cement-based grout as an alternative to conventional cement [9,15]. The aim is to create environmentally responsible, RAP-based highway material solutions offering a wide range of performance levels in terms of strength and stiffness. The term geopolymer usually refers to gels formed through alkaline liquid reacted with silica and alumina contained in alumina-silicates; in this case, sourced from by-product industrial wastes including fly ash (FA), ground generated blast furnace slag (GGBS), metakaolin (MK), and silica fume (SF). Use of these materials helps to offset the relatively high embodied carbon footprint of Portland cement or other types of bitumen or resin-based binder [24,25,26]. In this way, infusion of porous RAP with geopolymer grouts at ambient temperature offers an alternative type of waste-based pavement product. Related available literature considering mixtures of RAP and geopolymer grout without the use of heat or vibration for pavement applications is limited.

This paper initially characterises geopolymer grout performance in terms of flow time, setting time, and compressive strength. Use of selected grouts to infill voids in open-graded RAP skeletons to create SFC pavement materials is then explored, with performance evaluated based on permeable porosity, compressive strength, and ultrasonic pulse velocity test data. The microstructure of interfacial transition zones between RAP and geopolymer grout matrices is investigated using SEM observations. A key output from the reported research is a preliminary methodology to predict the stiffness of geopolymer-based SFC based on rapidly attainable laboratory or site-based test methods including strength and ultrasonic pulse velocity.

## 2. SFC Pavement Materials

SFC pavement specimens were manufactured at a laboratory scale using open-graded RAP aggregate skeletons infused with geopolymer grouts as explained in the following sections.

### 2.1. Open-Graded Aggregate Skeleton

Open-graded aggregate skeletons were prepared using 8–14 mm sized RAP particles with solid content levels ranging from 45–62% by volume. To achieve the 45% solid content level, RAP particles were placed in moulds without compaction. Otherwise, RAP skeletons were compacted manually to achieve the required solids content level. In a related study, open-graded aggregate skeletons with polymer modified emulsion binder were prepared using a vibrating compactor at 130 °C to achieve porosity levels ranging from 29–32% [9]. In contrast, both the un-compacted and compacted aggregate skeletons used in this study were prepared at room temperature and without the addition of any virgin bitumen or heating energy. The main properties of the RAP aggregates are presented in Table 1, together with an indication of the RAP skeleton preparation process in Figure 1a,b. While RAP bitumen content was not measured as part of this study, it was assumed to be within the range 5.8–6.3% [9,14]. RAP particles comprised original natural aggregate coated with irregular layers of aged bitumen as shown in Figure 1f. From subsequent SEM image analysis (see Figure 1g), interfacial transition zones (ITZ) between original aggregates and aged bitumen layers were largely porous in nature, with 10–40 µm diameter pores and 30–90 µm length fine cracks present; a significant feature given the established [27,28] impact of ITZ structure on the mechanical behaviour of cementitious materials.

### 2.2. Geopolymer Grouts

Geopolymers formed through reactions between an alkaline liquid activator and Si and Al contained in alumina-silicate-based binders were developed in this study using binders principally sourced as industrial by-products. Depending upon local resources and availability, solid alumina-silicate precursors can be in natural form such as zeolite, clays, shales, and amphibole or in industrial by-products such as fly ash (FA), ground-granulated blast furnace slag (GGBS), metakaolin (MK), silica fume (SF), red mud and waste glass [29]. In this study, the binders included fly ash, GGBS, silica fume, and metakaolin sourced locally from Kilroot power station (Carrickfergus, Northern Ireland), Ecocem Ireland Ltd. (Dublin, Ireland), Elkem (Hampshire, the UK) and Imerys (Cornwall, the UK) respectively. The chemical composition, particle size, and specific gravity of the materials are presented in Table 2 [29].

By considering diverse binders, the aim was to achieve a range of geopolymer grout properties. For instance, as high levels of grout flowability were potentially required, FA was considered based on its spherical shape and relatively smooth surface texture [30]. MK and SF were considered based on their reported contribution to good flow and high silicate and aluminium content [25], whilst GGBS was chosen based on its reported significant contribution to strength development without the need for heat curing [26]. Commercially available liquid activator, Geosil, with 45% solid potassium silicate (K_2_SiO_3_) content by mass, molar ratio of 1.6, and density of 1.51 g/cm^3^, was sourced from Woellner (Ludwigshafen, Germany) and used throughout for all geopolymer grout mixes.

## 3. Experimental Programme

### 3.1. Geopolymer Grout Mix Design

Table 3 is a mix design summary of the various geopolymer grout types considered. Investigated were binder combinations GGBS + FA, GGBS + FA + MK, and GGBS + FA + MK + SF with liquid-to-solid (LS) ratios ranging from 0.27–0.52. Based on previous related research [31], these binder combinations were chosen to offer a range of grout performance levels in terms of flow, setting time, and compressive strength, appropriate for a range of potential SFC pavement applications.

### 3.2. Geopolymer Grout Characterisation

Determined by measuring the time taken for 1200 mL of grout to flow through a Marsh flow cone apparatus with an internal orifice diameter of 12.7 mm, geopolymer grout flowability was assessed according to ASTM C939-02 [32]. It should be noted that grout fluidity is reported as being ideal for times in the range 8–35 s [20,32], albeit that these studies considered grout volumes of 1750 mL. Initial setting time of geopolymer grouts was defined by observing Vicat needle penetration according to BS EN 480-2:2006 [33]. Given geopolymer grout’s tendency to set more quickly than conventional portal cement grout, measurements in this study were recorded every 3–10 min (instead of 10 min as stated in the standard method) to improve accuracy levels. Compressive strength at 28 days was measured using 50 mm cubes according to BS EN 1015:11:1999 [34]. Specimens were covered with a polyethylene sheet and stored at room temperature at 20 °C until the time of testing.

### 3.3. SFC Characterisation

SFC samples were prepared by pouring geopolymer grout into moulds containing RAP skeletons from a height of around 30 cm to ensure full grout penetration (see Figure 1c,d). All SFC specimens were covered with polyethylene film and kept at room temperature until time of testing. For compressive strength measurements, 200 × 200 × 50 mm SFC slabs were initially cast, from which 50 mm cubes were cut using a diamond saw and discarding material from at least 15 mm from the slab edges (see Figure 1e). Testing was conducted using an ELE compression machine according to BS EN 1015:11 [34]. An average value of compressive strength was determined based on at least 3 specimens after 3, 7, and 28 days curing at room temperature.

Permeability of SFC specimens was determined by the vacuum saturation method according to ASTM 1202 [35]. This method was considered to be more accurate than alternative ASTM techniques such as cold-water and boiling water saturation [36]. Testing involved splitting 100 mm SFC cube specimens into two halves along the vertical plane with thin end layers removed to reduce edge effects. Specimen slices were then dried at 100 ± 10 °C for over 24 h, cooled at room temperature, and weighed to determine oven-dry mass ($W_{D})$. For each test specimen, three SFC slices were placed in a sealed desiccator connected to a vacuum pump operating at a pressure of −90 kPa and exposed to air drying for three hours followed by water saturation for a further one hour. The vacuum pump was then turned off and the specimens were soaked underwater in the desiccator for a further 20 h. Surface moisture was removed using a towel and specimens weighed to determine saturated mass ($W_{ST}$) and apparent mass in water ($W_{W}$). Permeable porosity ρ (%) of SFC specimens was then calculated using the equation:(1)$$\rho \left(\%\right)=\frac{W_{ST}-W_{D}}{W_{ST}-W_{W}}\times 100.$$

Ultrasonic pulse velocity (UPV) measurements were used to estimate material properties such as compressive strength and dynamic and static elastic moduli [37,38,39,40]. According to IS 13311 (Part 1):1992 [41], UPV can be used to classify concrete quality, with values in the range 3000–4500 m/s corresponding to a medium-good classification. In this study, 100 mm SFC cubes were assessed using a PUNDIT pulse velocity tester with 50 mm diameter transducers at 54 kHz based on BS EN 12504-4:2004 [42] using the equation:(2)$$UPV=\frac{\;L}{T},$$
where UPV is the ultrasonic pulse velocity (km/s); L is path length of the shortest distance from two transducers (mm); and T is transit time or the time spent by the ultrasonic pulse to transit through path length L (µs). Microstructural characteristics of RAP particles and SFC specimens were observed using SEM JEOL JSM-601PLUS apparatus (Hertfordshire, the UK). Except for RAP particles, all specimens with a dimension of approximately 15 × 15 × 12 mm were cut from SFC cubes using a diamond slicing wheel prior to sample preparation.

## 4. Results and Discussions

### 4.1. Geopolymer Grout Characterisation

In this phase of the research, all 20 of the GGBS + FA, GGBS + FA + MK, and GGBS + FA + MK + SF geopolymer grout mixes listed in Table 3 were characterised in terms of flow time, initial setting time, and 28-day compressive strength. Figure 2 illustrates the relationship between each property and LS ratio in the range 0.27 to 0.52. Given the diverse suite of mixes considered, a wide range of performance levels was achieved. To help categorise performance, flowability, initial setting time, and compressive strength results were classified as follows:Flow time (s): High (<24); Average (24–80); Low (>80);Setting time (mins): Fast (<25); Average (25–75); Slow (>75);28-day strength (MPa): Low (<40); Average (40–80); High (>80).


In terms of grout flowability (Figure 2a), water content was the clear dominant factor, with flow times generally decreasing with increasing LS ratio for all binder types considered. Very similar rates of ‘high’ performance were noted for all binder types at LS ratios greater than 0.38. Below 0.38, the influence of binder type became more significant, with a wide range of ‘average’ and ‘low’ performance levels noted; particularly at LS ratio 0.27. The GGBS + FA + MK binder exhibited the lowest level of flowability at this LS ratio, with a flow time of over 800 s. Looking forward to in situ application of this technology, grout flowability is a key property to control; particularly for large area grout pours into potentially well-compacted RAP. At the lowest LS ratio considered (0.27), the GGBS + FA + MK + SF binder combination offered the lowest flow time of 80 s (i.e., the highest flowability). In contrast to flow time, the LS ratio had a much less significant influence on grout setting time, particularly for LS ratios greater than 0.38 where performance levels attained steady state (Figure 2b).

Binder combination was the dominant controlling factor, with a wide disparity in setting times recorded across all LS ratios considered. For all binder combination types, setting time consistently decreased slightly at LS ratios less than 0.38. For all grout mixes exhibiting ‘high’ flowability, the corresponding range of setting times ranges from 27 (GGBS + FA binder) to 80 (GGBS + FA + MK + SF binder) min. Similar to flowability, grout setting time has practical significance when considering in situ applications. Whereas large area pours are likely to require ‘average’ or ‘slow’ setting times, smaller or emergency repair pours may require much shorter initial setting times. In this study, the fastest setting time was recorded for the GGBS + FA + MK + SF binder combination at a LS ratio of 0.27 (13 min). In terms of 28-day grout strength development (Figure 2c), the general trend for all binder types was increasing strength corresponding to decreasing LS ratio. Strength values increased dramatically when LS decreased from 0.52 to 0.27. Binder type had a significant influence on strength development, with values ranging from 56 MPa (GGBS + FA + MK) to 108 MPa (GGBS + FA + MK + SF) at the lowest LS ratio considered (0.27).

In summary, there is an element of performance contradiction. This was particularly the case for flow time and strength results, with mixes with the highest level of flowability (a characteristic likely to be deemed as favourable for large area pours) exhibiting the lowest values of strength, and vice versa. Within the ranges of performance levels of flow time, setting time, and compressive strength recorded, opportunity exists for selecting mixes with contrasting performance characteristics. This is highlighted by the solid and dashed lines added to Figure 2d, which demonstrate that for a starting design specification of ‘high’ flowability, for instance, mixes with ‘average’ setting time and either ‘average’ or ‘low’ strength can be chosen. Given the variation of pavement applications envisaged for this technology, this flexibility offers a significant benefit in terms of subsequent SFC implementation.

### 4.2. SFC Characterisation

The next phase of the research focused on exploring the impact of grout performance on the properties of resulting SFC specimens. From the 20 grout mixes previously described, four (labelled mix A, B, C, and D from this point forward) were chosen for this work as highlighted in Figure 2d and summarised in further detail in Table 4. Mixes A, B, and C were selected from the 40% GGBS +20% FA +20% MK +20% SF binder category and mix D from the 80% GGBS +20% FA category, based on the provision of contrasting performance classifications in terms of grout flowability, setting time, and compressive strength as follows:Mix A (‘High’|‘Slow’|‘Low’)Mix B (‘Average’|‘Average’|‘Average’)Mix C (‘Low’|‘Average’|‘High’)Mix D (‘Low’|‘Fast’|‘High’)


To develop a more comprehensive understanding of SFC behaviour, each of these grout types were then used to manufacture SFC test specimens comprising RAP skeletons with 45, 49, 54, and 62% solid contents by volume. Example images of resultant SFC specimens are provided in Figure 1d,e,i, as well as an SEM image of the aggregate-asphalt-geopolymer ITZ (grout mix B) in Figure 1h. In the latter, the visible aged bitumen layer is approximately 140 µm wide, with any non-visible localised pores and fine cracks filled/bounded by well-formed geopolymer grout. On further analysis of SEM images of this nature, networks of cracks with widths in the range 4–20 µm were evident in the ITZ between aged bitumen and geopolymer grout in the SFC specimens. This is a common mechanism reported in the literature [43] for materials incorporating cementitious- and bitumen-based materials. While this phenomenon may help to impede crack propagation in SFC materials and improve its energy-absorbing capacity [16,22,43,44], their presence will contribute to reduced levels of compressive strength.

Compressive strength results for the 16 SFC mixtures is presented in Figure 3, which shows wide ranges of performance at all ages. At 28-days for instance, and reflecting the wide range of mixture proportions considered, strength values ranged from 9 MPa (grout mix A, RAP content 62%) to 31.5 MPa (grout mix D, RAP content 45%). The 28-day compressive strength of SFC materials is in compliance with the recommended minimum compressive strength of 8 MPa for base layer established by the Design Manual for Roads and Bridges (DMRB): Volume 7–Section 2 [45] considering SFC as behaving similarly to hydraulically bound material (HBM) in accordance with BS 9227:2019 [46]. In terms of strength development with time, Figure 3a–d shows that, on average, SFC specimens gained approximately 80% of their 28-day strength value after three days. This trend reflects the established ability of geopolymer grouts to gain early strength rapidly [47] and offers a significant benefit for pavement applications where high early strength leading to early potential exposure to traffic is preferential. It is also clear from Figure 3 that there is a general negative influence of RAP addition on compressive strength. If considering geopolymer grout mix B for example, corresponding SFC strength at 28 days were 34, 32, 29, and 26% of the parent grout strength (67 MPa) as the RAP content increased from 45, 49, 54 to 62% respectively. Similar trends were noted for all SFC mixes, irrespective of the parent geopolymer grout type used (see Figure 3e–h).

SFC performance is further characterised in Figure 4d–f, which plots 28-day permeable porosity, ultrasonic pulse velocity, and compressive strength. In addition, plotted in Figure 4a–c are the properties of the parent geopolymer grouts used (mixes A, B, C, and D) in terms of their flow time, initial setting time and compressive strength. Key influences of both parent grout type and RAP addition on SFC performance can be reviewed simultaneously.

In terms of SFC compressive strength, significant influences of both RAP content (as highlighted in Figure 3) and parent grout strength are clear from Figure 4d, with increasing SFC strength corresponding to increasing grout strength and decreasing RAP contents respectively. It is clear from Figure 4d for SFC mixes comprising grout mix D there is an interrelated negative impact of grout flowability, initial setting time, and RAP content. Given the ‘low’ flowability of grout mix D (flow time > 600 s), full-depth aggregate skeleton penetration was not achievable at the higher RAP contents of 54 and 62% by mass; a problem compounded by mix D classified as having ‘fast’ setting set (13 min.) As a result, these SFC specimen types were deemed to have failed at the manufacturing stage (see Figure 4d) and further performance characterisation was not attempted.

In terms of permeable porosity, Figure 4e shows a less pronounced influence of RAP content when compared to compressive strength; particularly for grout types A and B (‘low’ and ‘average’ strength classifications respectively). For grout mixes C and D (‘high’ strength), a negative impact of increasing RAP content did emerge, albeit that performance levels were not possible for grout mix D at RAP contents 54 and 62%. The main factor influencing permeable porosity was the compressive strength of the parent grout used, with porosity values ultimately ranging from 20% for SFC specimens comprising grout mix A (36 MPa) to 11% for those comprising grout mix D (108 MPa).

In terms of ultrasonic pulse velocity, similar general trends were noted as for permeable porosity (see Figure 4f). Firstly, a minor influence of increasing RAP content was noted for SFC specimens comprising ‘low’ and ‘average’ strength grouts A and B. For ‘high’ strength grout mixes C and D, however, a clear influence emerged, with decreasing pulse velocities corresponding to increasing RAP contents. For example, the pulse velocity for grout mix C decreased from 4.1 to 3.6 km/s as the RAP content increased from 45 to 62% by mass. In addition, and reflecting improving paste microstructures, a general trend of increasing SFC pulse velocity with increasing grout strength is apparent in Figure 4f. Similar to permeable porosity, the lowest (3.3 km/s) and highest (4.4 km/s) values of pulse velocity were achieved by grout mixes A (36 MPa) and D (108 MPa) respectively. It is worth noting that, for conventional concrete, this range corresponds to performance quality category ‘Medium-Good’ as defined in IS 13311 (Part 1):1992 [41].

### 4.3. SFC Performance Predictions

Having undertaken the preliminary characterisation steps described above for SFC materials incorporating different types of geopolymer grouts and open-grade RAP skeletons, work progressed to review how the ultrasonic pulse velocity results might be utilised to provide meaningful rapid performance predictions. In the first instance, this was achieved by analysing the relationship between UPV and compressive strength for SFC; a relationship defined [37] by the exponential equation:(3)$$f_{cu}=a\cdot e^{\left(b\cdot UPV\right)},$$
where $f_{cu}$ is compressive strength (MPa); and a and b are empirical parameters determined by the least-squares method.

The relationship between UPV and compressive strength for the SFC results measured in this study are presented in Figure 5a, compared to published relationships for Portland cement concrete [40,48]. Comparable positive relationships between UPV and compressive strength exist for both SFC and conventional concrete, with the strongest relationship in Figure 5a associated with the SFC specimens assessed as part of this study (R^2^ = 0.87). Given this commonality, established relationships for conventional concrete in relation to elastic modulus (static and dynamic) were then compiled as shown in Figure 5b. This included using published relationships between elastic modulus and both UPV [38,49] and compressive strength [50,51]. With measured values from this study used as inputs into related prediction equations, comparable relationships existed for both approaches, with resulting values of static (*E_s_*) and dynamic (*E_d_*) elastic modulus ranging from 12–26 and 23–40 GPa respectively. As the work presented in this paper did not include direct measurement of SFC elastic modulus, this figure provided a means for deriving preliminary predictions of SFC elastic modulus based on measured values of UPV. As shown in Figure 5b, for instance, a measured UPV value of 4.0 km/s for SFC correlates to a predicted static elastic modulus value of 20 GPa.

### 4.4. Preliminary Design for Industrial Hardstanding Application

To investigate the practical implications of the work presented to this point, a preliminary design methodology for industrial hardstandings comprising SFC as a base layer is presented Figure 6. The approach adopted considers SFC as behaving similarly to a hydraulically bound material (HBM) in accordance with BS 9227:2019 [46].

Suitable materials included in this standard include cement, slag, and fly ash bound granular mixtures in accordance with BS EN 14227:2013 Parts 1–3 respectively [52,53,54], with permissible compressive strength classifications in the range C_.04/0.5_ to C_36/48_ (where the subscript figures define minimum values for cylinder specimens with a slenderness ratio of two and one, or cubes, respectively).

The 28-day strength value range for SFC recorded in this study (9–31 MPa) complies with this range and the minimum compressive strength of 8 MPa for base layer required by the Design Manual for Roads and Bridges (DMRB): Volume 7–Section 2 [45]. A simplified analytical pavement design approach presented by Williams [55] was used as the basis of the design methodology, which ignores the contribution of the surfacing and idealises the pavement as a two-layer system comprising HBM (or SFC in this case) on a supporting layer. The approach recognises that semi-flexible materials will ultimately crack under loading to form discrete slabs (not unlike paving concrete) and considers the stress situation at interior zones away from edges and corners. For the interior loading condition, the tensile stress (*s*) at the bottom of the HBM layer is given by the expression:(4)$$s=1.8p{\left(\frac{a}{h}\right)}^{1.85}\times \;log_{10}\left(\frac{E_{1}}{E_{2}}\right),$$
where *p* = tyre pressure; a = radius of tyre contact; *h* = layer thickness; *E*_1_ = layer modulus of elasticity; and *E*_2_ = foundation modulus of elasticity (approximated from 10 × CBR in MPa).

Equation (4) can be simplified by making use of the relationship between maximum wheel load (*P*) and tyre pressure (*p*) ($P=p\pi a^{2}$) and also by simplifying the power function from 1.85 to 2. As such, the equation may be rearranged to approximate the thickness of HBM layers as:(5)$$h={\left(0.57\left(\frac{P}{s}\right)\times log_{10}\left(\frac{E_{1}}{E_{2}}\right)\right)}^{0.5}.$$

The hardstanding surfacing layer, although ignored in the calculation, is assumed to compensate for edge/corner loading conditions that will induce cracks and produce greater stresses than the interior loading condition.

In the worked example presented in Figure 6, the assumed design inputs included: maximum wheel load, *P* (10 tonne, i.e., 100 kN); subgrade conditions (sand with CBR of 8%, i.e., *E*_2_ = 0.08 GPa); and pavement surfacing layer (80 mm asphalt layer). As shown in Figure 6a, the starting point of the design methodology required selection of a preferred SFC mixture. Selected in this instance was grout mix C with RAP volume of 62% and 28-day compressive strength of 16 MPa (correlating to strength class C_12/16_ in EN 14227:2013 [52]. This enabled subsequent tensile strength, UPV, and elastic modulus predictions of 1.9 MPa, 3.6 km/s, and 16 GPa respectively. Tensile strength prediction was based on relationships provided in BS EN 1992-1-1:2004 [56] for conventional Portland cement concrete, while for UPV and elastic modulus, the relationships presented previously in Figure 5 were used. Using Equation (5) above, this led to an SFC base layer thickness design of 265 mm.

## 5. Conclusions

The aim of this study is to investigate the properties of semi-flexible composite materials incorporating geopolymer grouts and reclaimed asphalt planings to develop innovative, predominantly waste-based pavement layers that do not require heating or mechanical compaction energy. Based on the results obtained, the following conclusions may be drawn:To facilitate the manufacture of SFC suitable for a broad range of practical applications, a diverse suite of 20 geopolymer grouts was initially produced using binder combinations GGBS + FA, GGBS + FA + MK, and GGBS + FA + MK + SF with liquid-to-solid (LS) ratios ranging from 0.27–0.52. The grouts had a wide range of performance in terms of flow (9–609 s), initial setting time (13–80 min), and compressive strength (19–108 MPa).A suite of 16 SFC mixtures was assessed based on four grout mixes chosen based on contrasting performance classifications. Each grout type was used to impregnate RAP skeletons with solids contents ranging from 45–62% by volume, resulting in corresponding wide ranges of SFC performance in terms of compressive strength (9–32 MPa), permeable porosity (10–20%), and ultrasonic pulse velocity (3.32–4.40 km/s). SFC performance was influenced by both grout properties and RAP content, with increasing performance values generally associated with decreasing RAP contents combined with highly flowable, high strength grout. All but two of the SFC mixtures considered, yielded viable pavement material solutions. Despite having the highest compressive strength (108 MPa), use of grout mix D was not practically possible with solid RAP contents of 54 and 62% by volume, owing to its relatively ‘slow’ flowability (609 s) and ‘fast’ setting time (13 min) resulting in incomplete RAP penetration.A strong correlation between ultrasonic pulse velocity and compressive strength was found for the range of SFCs considered (R^2^ = 0.87). Given the similarity between this relationship and those established for conventional Portland cement-based materials, published relationships relating UPV and elastic modulus for the latter were adopted to enable preliminary pavement designs incorporating SFC layers. An example for SFC use as an industrial hardstanding sub-base layer was presented. For a maximum wheel load of 10 tonnes, subgrade CBR of 8% and 80 mm-thick asphalt surfacing, the resultant SFC thickness requirement is 265 mm. For a hardstanding area of 100 m^2^, this equates to the consumption of approximately 35 tonnes of RAP and 15 tonnes of geopolymer-based product; thereby presenting a potentially economic and environmentally responsible pavement solution.The behaviour of SFC conformed with the mechanical performance levels required by the Design Manual for Roads and Bridges (DMRB): Volume 7–Section 2 [45] for base layer made of hydraulically bound material (HBM) in accordance with BS 9227:2019 [46]. As such, this initial investigation has successfully proven the potential suitability of this material.


## Figures and Tables

**Figure 1 materials-13-03644-f001:**
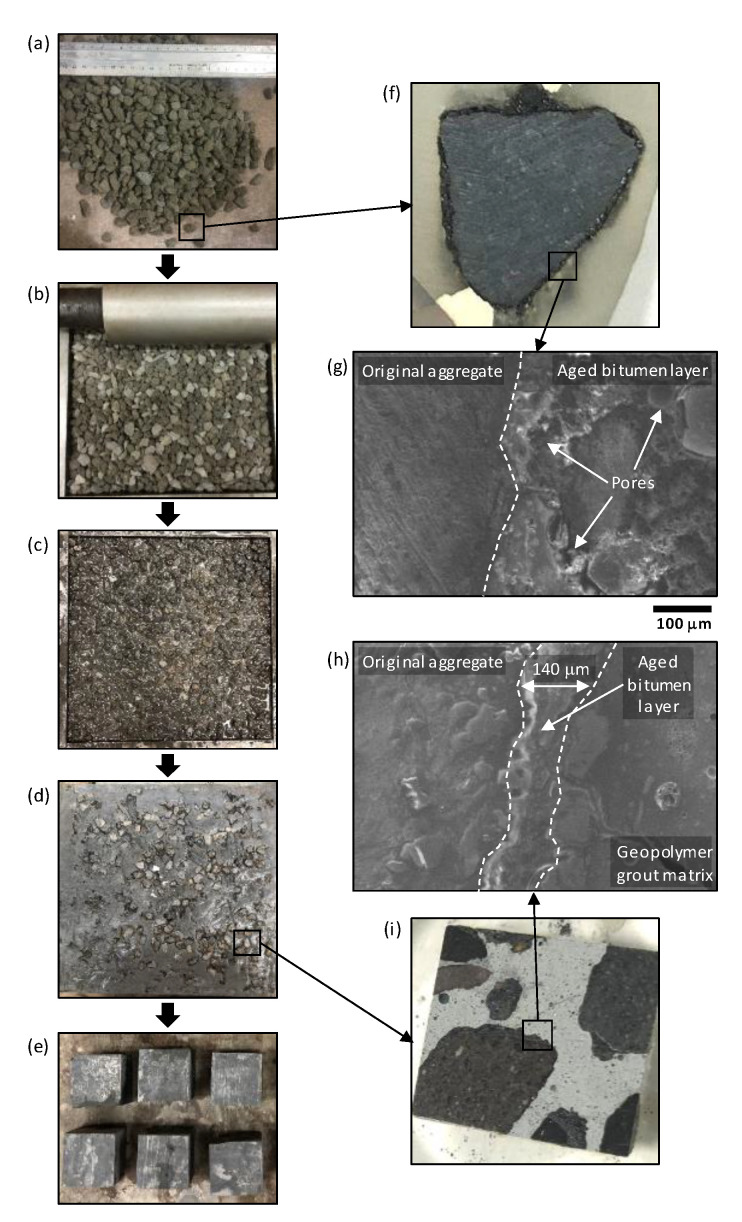
SFC manufacturing steps, including: (**a**) preparation of single-sized RAP particles; (**b**) hand compaction of RAP particles; (**c**) RAP particles infused with fresh geopolymer grout; (**d**) hardened SFC slab (200 × 200 × 50 mm); (**e**) extraction of SFC specimens for testing (50 mm cubes for compressive strength testing shown); (**f**,**g**) SEM characterisation of RAP particle; and (**h**,**i**) SFC specimen.

**Figure 2 materials-13-03644-f002:**
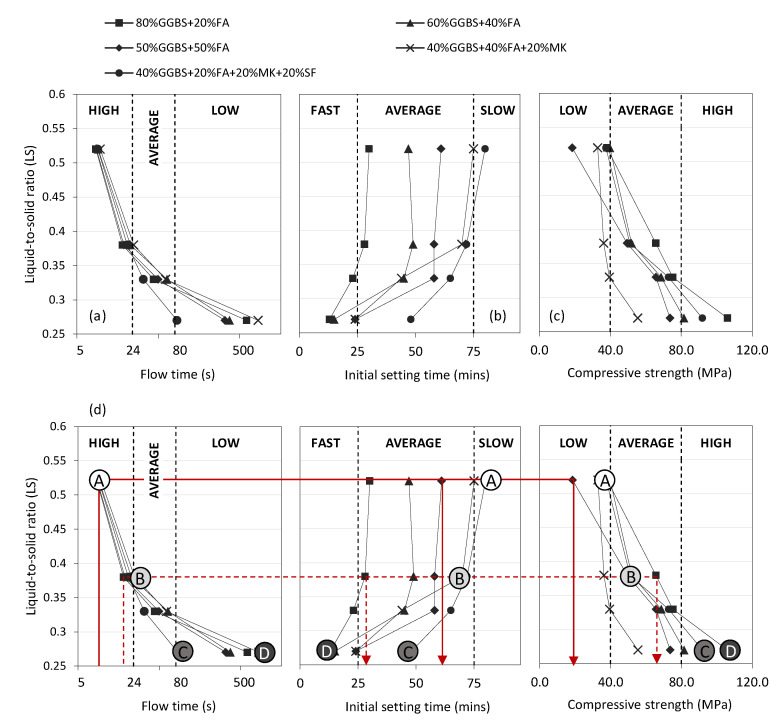
Performance of 20 geopolymer grout mixtures in terms of: (**a**) flow time; (**b**) initial setting time; and (**c**) 28-day compressive strength; (**d**) summary of selected grout mixes (Mix A, B, C, and D) for subsequent SFC characterisation phase.

**Figure 3 materials-13-03644-f003:**
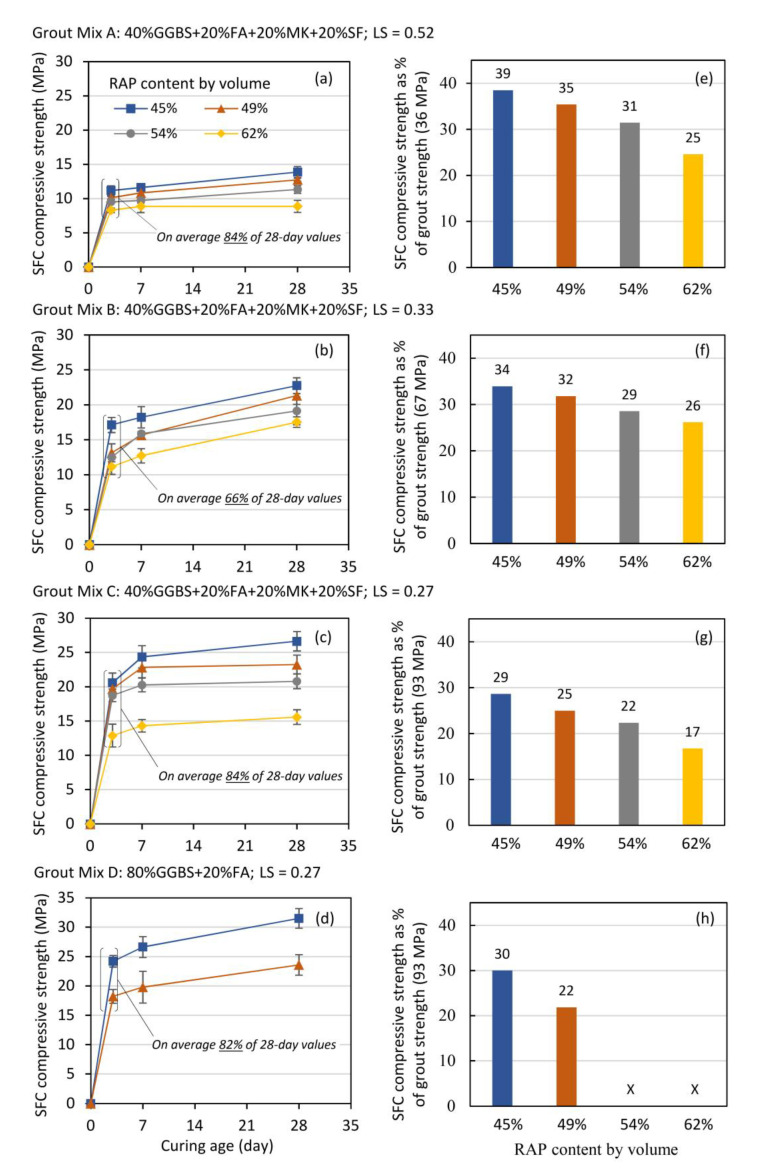
(**a**–**d**) SFC compressive strength development with time data; (**e**–**h**) 28-day SFC strength relative to the compressive strength of the parent grout at the same age.

**Figure 4 materials-13-03644-f004:**
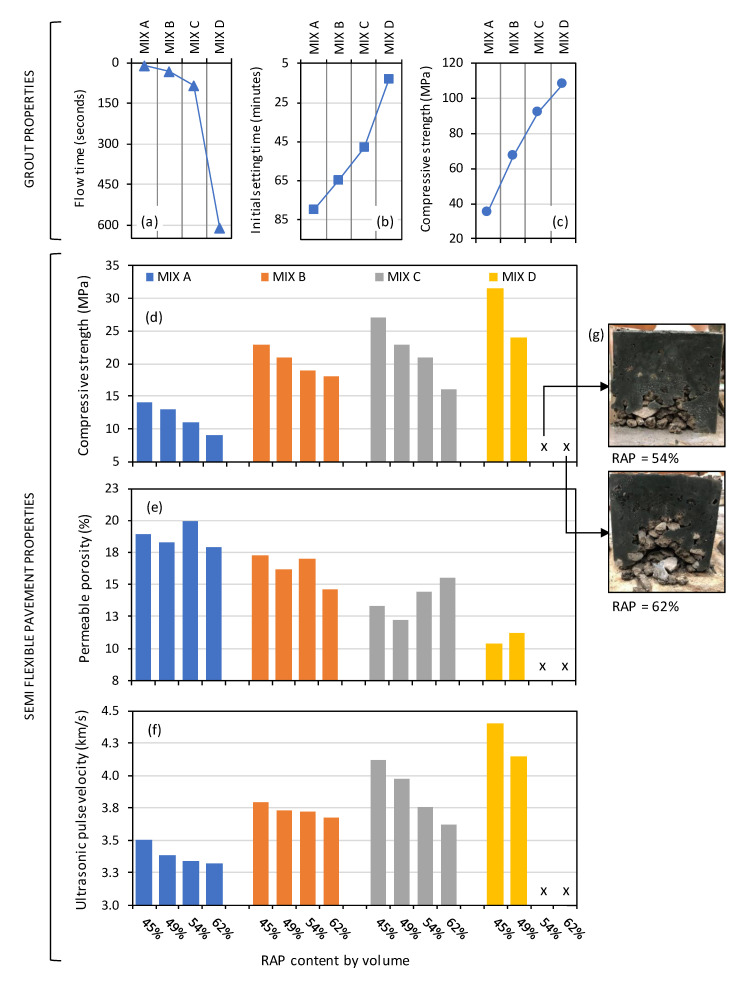
(**a**–**c**) Performance summary for grout Mix A, B, C, and D; SFC performance in terms of: (**d**) 28-day compressive strength; (**e**) permeable porosity; and (**f**) ultrasonic pulse velocity; and (**g**) images showing failure of selected specimens owing to insufficient grout penetration.

**Figure 5 materials-13-03644-f005:**
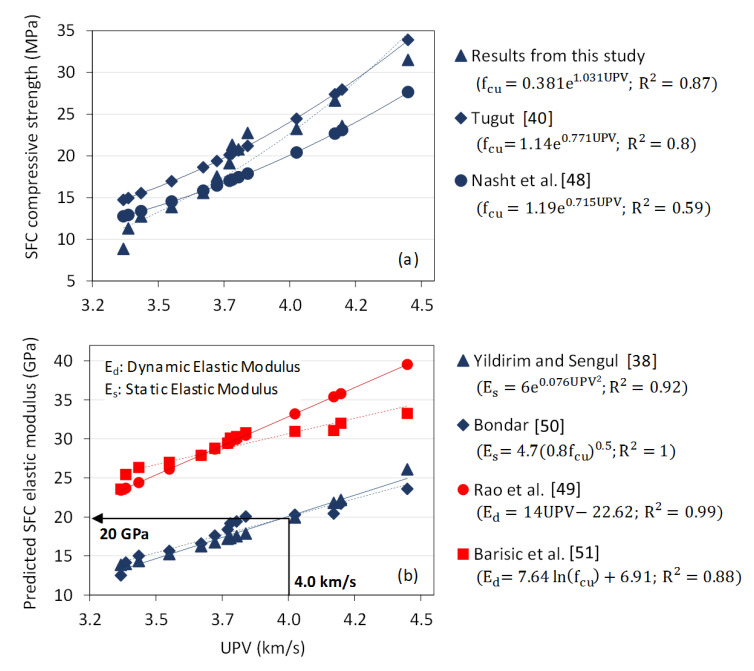
(**a**) Relationships between ultrasonic pulse velocity and compressive strength for both measured and published data (for Portland cement concrete); (**b**) Relationships between ultrasonic pulse velocity and both static and dynamic elastic modulus for published data (for Portland cement concrete).

**Figure 6 materials-13-03644-f006:**
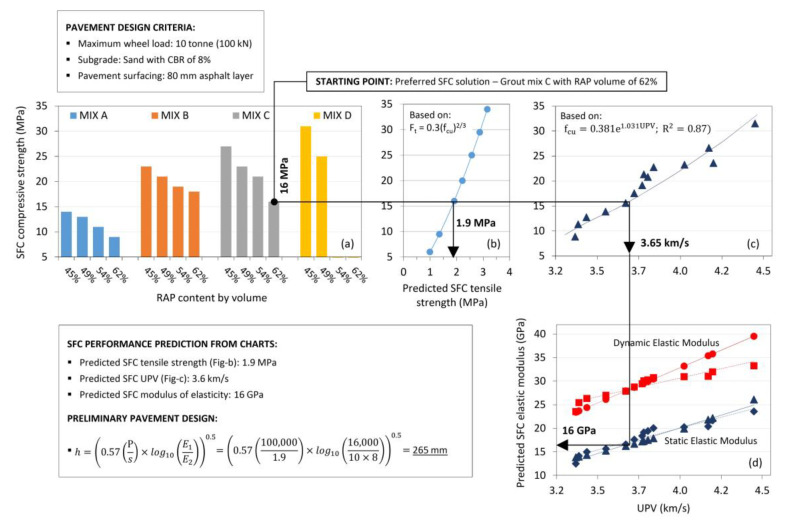
Mix design example for SFC utilised as a sub-base layer in a heavy-duty pavement application including: (**a**) laboratory-based compressive strength data; (**b**) predicted tensile strength values; (**c**) laboratory-based UPV data; and (**d**) predicted elastic modulus values.

**Table 1 materials-13-03644-t001:** Properties of RAP aggregate.

Properties	RAP
Compacted bulk density (g/cm^3^)	1.39
Loose bulk density (g/cm^3^)	1.25
Specific density (g/cm^3^)	2.53
Water absorption (%)	1.03
Moisture content (%)	0.31
Aggregate impact value (%)	5.10

**Table 2 materials-13-03644-t002:** Chemical composition, particle sizes, and specific gravity of geopolymer powders.

Material	Chemical Composition (% by Mass)				Particle Size ^1^ (μm)			Specific Gravity (g/cm^3^)
	SiO_2_	Al_2_O_3_	CaO	Fe_2_O_3_	D(10)	D(50)	D(90)	
FA	57	24	3.9	6	2.9	18.8	124.6	2.7
GGBS	36.5	10.4	42.4	0	1.1	5.3	22.5	2.85
MK	55	40	0.3	1.4	0.9	2.7	8.2	2.6
SF	96	0.8	0.5	0.8	0.1	0.15	0.4	2.2

^1^ where D(10), D(50), and D(90) are 10%, 50%, and 90% of particles smaller than this size respectively.

**Table 3 materials-13-03644-t003:** Geopolymer grout compositions.

Binder Combinations	Geopolymer Powder Contents (% by Mass of Total Binder)				Liquid-to-Solid Ratios (LS)
	GGBS	FA	MK	SF	
GGBS + FA	80	20	0	0	0.27, 0.33, 0.38, 0.52
	60	40	0	0	
	50	50	0	0	
GGBS + FA + MK	40	40	20	0	
GGBS + FA + MK + SF	40	20	20	20	

**Table 4 materials-13-03644-t004:** Properties and composition of geopolymer grouts used for SFC pavement material.

MIX	GGBS/FA/MK/SF Binder Composition (%)	LS	Grout Properties			Grout Performance Summary: Flowability|Setting Time|Strength ^1^
			Flow (s)	Setting Time (mins)	Strength (MPa)	
A	40/20/20/20	0.52	9.0	80	36.0	*‘High’|‘Slow’|‘Low’*
B	40/20/20/20	0.33	32.6	65	67.0	*‘Average’|‘Average’|‘Average’*
C	40/20/20/20	0.27	84.8	48	93.0	*‘Low’|‘Average’|‘High’*
D	80/20/0/0	0.27	608.6	13	108.0	*‘Low’|‘Fast’|‘High’*

^1^ Grout performance summary classification:
**Flow Time (s):****>80****24–80****<24**Flowability:‘*Low*’‘*Average*’‘*High*’Initial setting time (mins):>7525–75<25Setting time:‘*Slow*’‘*Average*’‘Fast’28-day compressive strength (MPa):<4040–80>80Strength:‘*Low*’‘*Average*’‘*High*’

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
