# Peer review of "A Preliminary Characterisation of Innovative Semi-Flexible Composite Pavement Comprising Geopolymer Grout and Reclaimed Asphalt Planings"

_materials, 2020, doi:10.3390/ma13163644_

Round 1

Reviewer 1 Report

This paper introduces the development process of a semi-flexible pavement material made with reclaimed asphalt planings (RAP) and geopolymer cement-based grouts using industrial by products to offer economic and environmental savings relative to conventional Portland cement-based grouting systems. The topic and content of the study are significant for environmental protection and sustainable development of highway transportation infrastructure. The preliminary research conclusions are of reference value for similar projects. However, according to the experience of semi-flexible pavement materials, cracking should be the main type of distress of this kind of materials. In this sense, in addition to compressive strength, crack resistance and durability of materials should also be the main consideration of material design.

Author Response

Special Issue "Novel Materials and Technologies for the Urban Roads of the Future"
Materials (ISSN 1996-1944; CODEN: MATEG9)
Manuscript ID: materials-853453
Dear Editors in Materials journal,
Thank you for giving us the opportunity to submit a revised manuscript “A preliminary characterisation of innovative semi-flexible composite pavement comprising geopolymer grout and reclaimed asphalt planning” for publication in the special issue "Novel Materials and Technologies for the Urban Roads of the Future" of Materials journal.
We appreciate the time and effort that you and the reviewers dedicated to providing and helpful comments for correction or modification. The manuscript has been revised to address the reviewer comments, which are appended alongside our responses to this letter. All page numbers refer to the revised manuscript file with tracked changes.
Sincerely,
The authors: An Huynh, Dr. Bryan Magee, Dr. David Woodward
Reviewers' Comments to the Authors:
Reviewer 1
“This paper introduces the development process of a semi-flexible pavement material made with reclaimed asphalt planings (RAP) and geopolymer cement-based grouts using industrial by products to offer economic and environmental savings relative to conventional Portland cement-based grouting systems. The topic and content of the study are significant for environmental protection and sustainable development of highway transportation infrastructure. The preliminary research conclusions are of reference value for similar projects. However, according to the experience of semi-flexible pavement materials, cracking should be the main type of distress of this kind of materials. In this sense, in addition to compressive strength, crack resistance and durability of materials should also be the main consideration of material design.”
Author response: We agree with the reviewer’s suggestion, it would have been interesting to explore the aspects of crack resistance and durability of materials. Since this study focuses on investigating the mechanical properties of SFC materials such as compressive strength and elastic modulus, other factors of material design will be discussed in detail in future works. Work in a subsequent paper will discuss the durability and crack propagation of SFC pavements before and after subjected to simulated trafficking by road test machine.

Reviewer 2 Report

this manuscript provides a preliminary study of the semi-flexible composite pavement with RAP, generally the findings are useful to audiants in this field. the authors may need to highlight the innovations of this study from existing studies related to semi-flexible pavement, as there are already quite a few similar studies. below are a little detailed comments;

  1. reclained asphalt planing, is this the same are reclaimed asphalt pavement?
  2. Fig.3, how may replicates were used for the bar chart? may need to add the error bar. 
  3. what's the properties of the RAP?e.g.,the aggregate size, asphalt content?
  4. fig.5, the authors cited four different studies from others? I am wondering if this correlation universal? is the correlationship material dependent? e.g., the cement type, aggregate gradation would bring difference for the curve?  

Author Response

“This manuscript provides a preliminary study of the semi-flexible composite pavement with RAP, generally the findings are useful to audiants in this field. the authors may need to highlight the innovations of this study from existing studies related to semi-flexible pavement, as there are already quite a few similar studies. below are a little detailed comments;”
1. Reclaimed asphalt planing, is this the same as reclaimed asphalt pavement?
Author response: The term “Reclaimed asphalt planing” is the same as “reclaimed asphalt pavement”. It means the materials are removed from asphalt pavement by planing operations [1].
2. In Fig.3, how many replicates were used for the bar chart? May need to add the error bar.
Author response: Figure 3 shows compressive strength results for SFC mixture at 3, 7 and 28 days. The values of 28-day compressive strength of SFC materials in Figure 3 (a-d) were re-used in Figure 3 (e-h) to highlight the negative influence of RAP addition on SFC compressive strength. Each result published in these figures is the average of three values.
Thank you for your suggestion. Error bars were added in Figure 3 (a-d) in lines 312-313 on page 10.
3. What's the properties of the RAP? e.g., the aggregate size, asphalt content?
Author response: The properties of RAP particles such as densities, water absorption, moisture content and aggregate impact value are presented in Table 1. As stated in line 87, the size of RAP used in this study is 8-14 mm. For SFC materials in this study, RAP was used as a substitute for conventional aggregate in cold mix so aged bitumen content was assumed to be in a stiff form attached to old aggregate. As such, the old aggregate and asphalt content of RAP was not considered in this study. However, we have added the assumed range of bitumen content obtained from previous papers working on similar type of RAP [2,3] in lines 96-97 on page 3.
4. In Fig.5, the authors cited four different studies from others? I am wondering if this correlation universal? Is the correlationship material dependent? e.g., the cement type, aggregate gradation would bring difference for the curve?
Author response: Thank you for pointing this out. Figure 5 (b) presents the relationships between ultrasonic pulse velocity (UPV) and predicted SFC elastic modulus. Four different studies cited includes Yildirim and Sengul [38], Bondar [48], Rao et al. [47] and Barisic et al. [49]. Due to the lack of available information of established relationship of UPV-elastic modulus for SFC materials, the correlation of UPV and elastic modulus from mentioned studies for conventional concrete was used in this study. A comparable positive relationships between UPV and compressive strength exist for both SFC and conventional concrete (Figure 5 (a)) which proves their commonality of these correlations. Since the correlations are possibly influenced by various factors such as water/cement ratio, cement types, aggregate gradation and so on, various established relationships of UPV-elastic modulus and compressive strength-elastic modulus were used to provide evidence supporting of the comparable positive relationships among mentioned mechanical properties.

Reviewer 3 Report

The article is almost complete and well written. Some information must be added before publication for a better appreciation of the article. 

  • In Table 1, what is the bitumen content of the RAP.
  • In section 2.2, more information is needed on the alkaline activator.
  • The author must comment if these industrial by-products are easily available. There is a general trend that these materials are harder to find and it would be good to know the point of view of the authors.
  • In 4.2, it would be good to know more about the target compressible strength and permeable porosity. 
  • In 4.3, recall what is SFC at the start of the section.
  • In 4.4., it would be good to recall the figure number that shows that the materials fit the compressive strength classification. The classification code can be further explained.
  • In conclusion, you can add a point regarding the normative aspects.

Author Response

“The article is almost complete and well written. Some information must be added before publication for a better appreciation of the article.”
1. In Table 1, what is the bitumen content of the RAP?
Author response: RAP particles were composed of old aggregate, bitumen layers and/or cluster of fine aggregates which would be an issue if RAP was re-used in hot-mix asphalt because aged bitumen content would be liquefied during heating process and it would scatter into finer particles. However, for SFC materials in this study, RAP was used as a substitute for conventional aggregate in cold mix so aged bitumen content was assumed to be in a stiff form attached to old aggregate. As such, the old aggregate and asphalt content of RAP was not considered in this study. However, we have added the assumed range of bitumen content obtained from previous papers working on similar type of RAP [2,3] in lines 95-96 on page 3.
2. In section 2.2, more information is needed on the alkaline activator.
Author response: As suggested by the reviewer, more information including molar ratio, density and source of alkaline activator was added in Section 2.2 in lines 123 - 125 on page 4.
3. The author must comment if these industrial by-products are easily available. There is a general trend that these materials are harder to find and it would be good to know the point of view of the authors.
Author response: As suggested by the reviewer, we have added the availability aspects of solid alumina-silicate precursors sourced from industrial by-products in section 2.2 in lines 106 - 109 on page 3.
4. In section 4.2, it would be good to know more about the target compressible strength and permeable porosity.
Author response: As suggested by the reviewer, minimum compressive strength value for base layer required by the Design Manual for Roads and Bridges (DMRB): Volume 7 – Section 2 was added as the target compressive strength in Section 4.2 in lines 266 - 270 on page 8. In terms of the target for permeable porosity, although we agree that this is an interesting consideration, the main focus of this study is still the mechanical properties of SFC for the pavement design. As such, the target for permeable porosity would be considered in future works.
5. In 4.3, recall what is SFC at the start of the section.
Author response: We have added the suggested content to the manuscript at the start of section 4.3 in lines 323 – 324 on page 12.
6. In 4.4., it would be good to recall the figure number that shows that the materials fit the compressive strength classification. The classification code can be further explained.
Author response: As suggested by the reviewer, the compressive strength requirements for base layer was recalled in section 4.4 in lines 356 – 358 on page 12.
7. In conclusion, you can add a point regarding the normative aspects.
Author response: As suggested by the reviewer, a normative conclusion was added in section 4.4 in lines 425 – 426 on page 14.
References
[1] Carswell, I.; Nicholls, J.C.; Widdyatmoko, I.; Harris, J.; Taylor, R. Best practice guide for recycling into surface course. Road Note. Transport Research Laboratory. 2010.
2
[2] M. L. Afonso, M. Dinis-Almeida, L. A. Pereira-De-Oliveira, J. Castro-Gomes, and S. E. Zoorob, “Development of a semi-flexible heavy duty pavement surfacing incorporating recycled and waste aggregates - Preliminary study,” Constr. Build. 697 Mater., vol. 102, pp. 155–161, 2016.
[3] N. Hossiney, G. Wang, M. Tia, and M. Bergin, “Evaluation of concrete containing reclaimed asphalt pavements for use in concrete pavement,” Transportation Research Record, pp. 1–13, 2008.
